# Mechanistic Investigation of the Androgen Receptor DNA-Binding Domain and Modulation via Direct Interactions with DNA Abasic Sites: Understanding the Mechanisms Involved in Castration-Resistant Prostate Cancer

**DOI:** 10.3390/ijms24021270

**Published:** 2023-01-09

**Authors:** Shangze Xu, Matthew D. Kondal, Ayaz Ahmad, Ruidi Zhu, Lanyu Fan, Piotr Zaborniak, Katrina S. Madden, João V. de Souza, Agnieszka K. Bronowska

**Affiliations:** 1Chemistry—School of Natural and Environmental Sciences, Newcastle University, Newcastle Upon Tyne NE1 7RU, UK; 2Newcastle University Centre for Cancer, Newcastle University, Newcastle Upon Tyne NE1 7RU, UK; 3School of Engineering, Newcastle University, Newcastle Upon Tyne NE1 7RU, UK; 4Translational and Clinical Research Institute, Newcastle University, Newcastle Upon Tyne NE2 4HH, UK

**Keywords:** androgen receptor, castration-resistant prostate cancer, dimerisation, molecular dynamics simulations, protein-protein interactions, DNA-protein interactions, DNA-protein adducts

## Abstract

The androgen receptor (AR) is an important drug target in prostate cancer and a driver of castration-resistant prostate cancer (CRPC). A significant challenge in designing effective drugs lies in targeting constitutively active AR variants and, most importantly, nearly all AR variants lacking the ligand-binding domain (LBD). Recent findings show that an AR’s constitutive activity may occur in the presence of somatic DNA mutations within non-coding regions, but the role of these mutations remains elusive. The discovery of new drugs targeting CRPC is hampered by the limited molecular understanding of how AR binds mutated DNA sequences, frequently observed in prostate cancer, and how mutations within the protein and DNA regulate AR-DNA interactions. Using atomistic molecular dynamics (MD) simulations and quantum mechanical calculations, we focused our efforts on (i) rationalising the role of several activating DBD mutations linked to prostate cancer, and (ii) DBD interactions in the presence of abasic DNA lesions, which frequently occur in CRPC. Our results elucidate the role of mutations within DBD through their modulation of the intrinsic dynamics of the DBD-DNA ternary complex. Furthermore, our results indicate that the DNA apurinic lesions occurring in the androgen-responsive element (ARE) enhance direct AR-DNA interactions and stabilise the DBD homodimerisation interface. Moreover, our results strongly suggest that those abasic lesions may form reversible covalent crosslinks between DNA and lysine residues of an AR via a Schiff base. In addition to providing an atomistic model explaining how protein mutations within the AR DNA-binding domain affect AR dimerisation and AR-DNA interactions, our findings provide insight into how somatic mutations occurring in DNA non-coding regions may activate ARs. These mutations are frequently observed in prostate cancer and may contribute to disease progression by enhancing direct AR-DNA interactions.

## 1. Introduction

Prostate cancer is currently a prevalent disease, the second most common cause of cancer death in the UK and the USA [1,2,3,4,5]. The androgen receptor (AR), a hormone-inducible transcription factor, represents an important therapeutic target in prostate cancer. The AR is activated by steroid recruitment to its ligand binding domain (LBD), followed by receptor nuclear translocation and dimerisation via the DNA binding domain (DBD) [6]. Clinically used small-molecule drugs bind to the LBD and interfere with steroid recruitment but are rendered ineffective by expressing constitutively active variants, such as ARV7, that lack the LBD [7,8,9,10]. Also, point mutations in the AR ligand-binding domain (LBD) have been identified that confer resistance to abiraterone and enzalutamide [11]. Likewise, somatic mutations in the DBD are linked to treatment resistance [12,13]. These drug-resistance mechanisms confound treatment of this ‘castration resistant’ stage of prostate cancer, characterised by the return of AR signalling. Despite the approval of several new therapeutics to treat metastatic prostate cancer following the development of castration resistance, the disease remains an unmet clinical need [14].

To date, several gain-of-function mutations have been found in the DBD and linked to prostate cancer, such as K581R [15,16,17]. Other mutations, such as L575P, A587V and A588S, were found in prostate cancer, but their specific roles remain unknown [18]. Except K581, those residues are located far from the dimerisation interface, are not involved in direct interactions with DNA, and their exact role is yet to be elucidated. While mutations in the AR, particularly within the LBD domain, have been extensively studied in past years, little has been reported about somatic DNA mutations at the non-coding regions where DBD binds to DNA. Lack and coworkers have addressed this gap, employing clinical whole-genome sequencing [18]. They showed that DBD binding sites have a dramatically increased rate of mutations that is greater than any other transcription factor and specific to prostate cancer [18]. They also provided evidence that these mutations at DBD-DNA binding sites are caused by impaired repair of abasic sites [19]. Most AR-DBD-binding sites are in intronic or intergenic regions. Many of them contain an androgen response element (ARE) that consists of a 15-bp palindromic sequence 5′-AGAACA-NNN-TGTTCT-3′ or two hexameric direct repeat sequences 5′-AGAACA-NNN-AGAACA-3′ with a 3 bp spacer [20]. However, the impact of those abasic lesions on the structure and dynamics of the AR-DNA complex has not been studied to date.

To consolidate results on mutations within DBD which interact with androgen-responsive elements (AREs) in DNA, we performed a computational study employing a combination of atomistic molecular dynamics (MD) simulations, solvent mapping, and quantum mechanical (QM) approaches. First, through atomistic MD simulations, we rationalised the role of several activating DBD mutations linked to prostate cancer: L575P, K581R, A587V, and A588S. We showed that these mutations modulate AR through conformational changes, altering both interfaces and increasing DNA interactions. Next, we focused on DBD-DNA interactions in the presence of apurinic DNA lesions within the androgen-responsive element (ARE), which frequently occur in CRPC. We report microsecond timescale all-atom MD simulations that capture the conformational dynamics of ARE bearing apurinic lesions in the presence of dimerised DBD. We found that these lesions enhance direct AR-DNA interactions by locally changing the ternary complex’s structure and dynamics, stabilising the DBD homodimerisation interface. Moreover, our results strongly suggest that those abasic lesions may form reversible covalent crosslinks between DNA and lysine residues of AR via Schiff base formation. Our MD simulations identify and quantify the frequency of several spontaneous, non-covalent interactions between the lesions and lysine residues from DBD that prefigure AR-DNA covalent crosslinks. We then followed up with an investigation of those crosslinks at the quantum mechanical level of theory, focusing on their structure and molecular orbitals. Such multiscale computational mapping of DBD-DNA crosslinks gives essential insights for further experimental studies involving site-directed mutagenesis and mass-spectrometry studies to confirm the presence of these crosslinks in CRPC.

## 2. Results

### 2.1. Intrinsic Molecular Dynamics of AR-DBD Homodimers in the Presence and Absence of DNA

To investigate the monomer structural stability and homodimerisation, all-atom molecular dynamics (MD) simulations ensembles were obtained for the monomer, dimer unbound to DNA, and the ternary complex: AR-DBD homodimer bound to DNA androgen response element (ARE).

The internal organisation of the monomer changed from a structured cross-helix-loop-helix motif to a partially disordered conformation, as shown in Figure 1A. The monomer adopted the partially disordered conformation by a transitional configuration with both helix1 and helix3 antiparallel to each other, disrupting the necessary conformation to bind to DNA. This conformational change occurred in the first half of the simulation, as shown in the root mean-squared deviation (RMSD) changes in Figure 1B. The two cysteine-Zn motifs moved away from each other, as shown in Figure 1C. The Appendix A shows the distance between the two Zn^2+^ ions in each monomer, showing how AR-DBD monomers unfolded through time in the absence of DNA.

The homodimer simulation showed that the dimerisation interface was unstable without DNA. The DBD homodimer quickly disassembled through the simulations, as represented by the centre-of-mass distance between monomers shown in Appendix A. This event occurred within the first 50 ns of the simulation, with interfacial interactions being weakened throughout the simulated time. In contrast, the protein-protein interaction (PPI) interface was maintained throughout the simulation time when the system was modelled with the DNA ARE bound to the protein homodimer. As shown in Figure 2A, the system’s average conformation after 500 ns showed interactions between both chain D-box regions, maintaining the internal structural cohesion. The main interactions between the dimers’ D-boxes came from the hydrogen bonds between R599 and D601, as shown in Figure 2B.

The stabilising interactions between androgen responsive element (ARE) DNA and AR-DBD interactions were maintained throughout the simulations. This included interactions between the DNA backbone and a series of lysines (K610 and K581), arginines (R586 and R616) and serines (S579) located at the DNA interface, especially along the P-box region. These residues interact with specific bases (such as interaction between R586 and guanine G30) via hydrogen bonds. The overall complex was stabilised via salt bridges between lysines and the backbone phosphates. Hence, the regions which included these residues showed low root mean square fluctuations (Figure 2B).

Arginine R586 is the residue that sampled the most stable hydrogen bonds (quantified by occupancy in sampled frames) throughout the simulation replicas. These interactions involved R586 side chain (acting as a donor) and guanosine G13 as acceptor of both DNA chains. The highest occupancy of these hydrogen bonds is 32%. Another crucial interaction between DNA and AR-DBD involved R616 (highest occupancy of 40%) and S579 (highest occupancy of 50%).

### 2.2. Impact of Cancer-Linked Mutations within the AR-DBD

To study the protein point mutations that would affect DNA-DBD interactions, we selected four clinically relevant point mutations and investigated these using all-atom molecular dynamics simulations: L575P, K581R, A587V, and A588S. The monomer simulations of the four mutations showed that they are prone to unfold similarly to the WT monomer. This is evidenced by differences in distances between the two coordinated Zn^2+^ within the chain, as shown in Appendix A. However, there were differences in the behaviour and interactions between the mutated AR-DBD and DNA.

The L575P mutation did not show different overall dynamics to the WT, evidenced by low RMSD values (Figure 3B). The L575P mutation, however, affected the N-terminal region of the DBD domain. The antiparallel β-sheet located in the N-terminus underwent a substantial conformational change, which allowed helix2 to be buried into the major groove (Figure 3A).

The mutations A587V and A588S are located at the end of helix2. The A587V mutation affected the region surrounding the 587 positions by increasing the volume of the side chain. The bulkier valine sidechain pushed the Y594 from a buried position to a conformation that allowed the hydroxyl group to assemble a hydrogen bond with two residues located within another monomer: the N611 side chain, and the backbone of K610. In comparison, in wild-type simulations, Y587 moved towards the outside region of the protein, becoming exposed to the solvent, without generating specific interactions, as shown in Figure 3A.

The A588S mutation changed the overall electrostatic characteristics of the helix2 terminal region. This affected the behaviour of the androgen responsive element and its interaction with the DBD. The S588 maintained the structure of the helix2 via a hydrogen bond between the S588 hydroxyl group and the backbone of F594. This hydrogen bond was sampled in two out of the three simulation replicas. In the replica wherein this hydrogen bond was not sampled, the S588 interacted with the backbone of K610 and phosphate oxygen atoms of ARE’s backbone.

In contrast to the apolar mutations described above, the A588S mutation induced a lateral movement of the dimer. As shown in Figure 4B, the dimer interface containing the A588S mutation partially unbound from its protein partner and its DNA element. This resulted in a twisting movement that separated both monomers. This motion is related to an increase in the number of contact points between the protein and DNA. This can be quantified by the increase in the RMSD, and an increase in protein-DNA hydrogen bonds (Figure 4C,D, respectively).

The K581R mutation showed similar dynamics to the A588S mutation. The dimerisation twist movement was sampled in all three replicas, with the R581 shifting from the major groove, where helix2 and R581 are located, as shown in Figure 5. Also, the P-box region, where the R581 is sampled, unbound from the major groove, weakening the interactions between DNA and protein.

### 2.3. Impact of Abasic Lesions within ARE on Interactions with AR-DBD Domain

To study how the depurination of the nucleotides within ARE would affect AR-DBD binding and intrinsic dynamics, we generated six models with basic substituents, comprising cytidine C11-5′-3′ depurinated with the hydroxyfuranose ring, cytidine C11-5′-3′ depurinated with the ester-furanose ring, and cytidine C11-5′-3′ depurinated with the furanose ring-opened. The remaining three models were the same transformations on cytidine C11-3′-5′ in the reverse strand.

The dynamics of the ester-furanose directly affected the backbone dynamics of the ARE element. In both simulated configurations, the ester-furanose rings twisted outwards from the DNA. The ester-furanose twisted its backbone for the leading strand, and the ester group buried itself in the helix located between F583–A587. This torsion movement created a pocket that allowed adenosine A12 to occupy the region where the ester-furanose was located. For the counter-strand, a similar motion occurred. However, the ester group faced the solvent instead, as shown in Figure 6A.

The abasic site in a hydroxyfuranose variant did not interact directly with the AR-DBD domain. The hydroxyfuranose position 11 interacted as a hydrogen bond donor between its hydroxyl group and the previous nucleic base. This interaction generated a kink in the DNA, but this structural change was smaller than the conformational change caused by the ester-furanose.

The aldehyde group within the opened furanose ring positionally fluctuated more than the previous configuration for both 11 and 28 positions, as shown in RMSF Figure 6B. Still, both positions interacted directly with the nearest K610. As shown in Figure 6C, the aldehyde oxygen interacted with the NZ atom within the side chain of the K610, with several sampled conformations showing interatomic distances below 0.35 nm, indicating an interaction between these atoms. This feature was more prominent in position 28 than in position 11, which required longer timescales to obtain similar configurations.

Although the DNA’s RMSD did not differ significantly between lesioned strands and WT, the total number of hydrogen bonds for the open furanose ring in position C11 3′-5′ (the average number of bonds = 8 ± 1) was the highest for the set of abasic lesions simulated. The protein conformational changes caused by the substitution to an abasic mutant was not directly related to the specific protein-DNA interactions. As shown in Figure 7B,C, the RMSD of the protein and of the DNA showed a clear correlation with the changes in hydrogen bond numbers between DNA and protein.

Given this stable DNA-protein interaction in the open furanose ring at position 11 of the leading strand, we calculated the molecular orbitals of the lesion area using density functional theory (DFT). As shown in Figure 8, the highest-occupied molecular orbital (HOMO) is located primarily at the aldehyde group, and the lowest-unoccupied molecular orbital (LUMO) is located entirely at the side chain amine group of the K610. Moreover, the electron density of the frontier bonding molecular orbital is located between the aldehyde group and the amine, as shown in Figure 8C.

## 3. Discussion

Understanding the AR-DBD domain’s assembly may open new avenues to target the dimerisation process and its interaction with AREs with small molecules, to yield future therapeutics. Our studies have shown that the structure of the complex composed by the AR-DBD homodimer bound to DNA duplex is significantly more stable than the DNA-free AR-DBD homodimer. In all-atom MD simulations, the disassembly of AR-DBD dimers occurred in less than 50 ns for all three simulated replicas, indicating how rapidly and easily the disassembly events may occur. To the contrary, the interaction interface of the wild-type homodimer bound to DNA did not go through any major conformational changes during the simulations. This observation agrees with the findings by Helabad and colleagues [21], which showed that the androgen receptor ternary complex bound to DNA is highly stable and does not disassemble spontaneously.

The results of our simulations have also shown that the wild-type AR-DBD monomer went through significant conformational changes, which affected both the D-box and P-box within the DBD domain. These structural fluctuations indicated that the assembly of androgen receptor-DNA complex is likely to occur stepwise: the homodimerisation of the protein induces the conformational changes required for high-affinity binding to ARE, and the protein dimer is stabilised by the DNA interactions. It is important to note that our simulations focused solely on the DBD domain of the androgen receptor, therefore the impact of N-terminal (NTD) and ligand-binding (LBD) domains on the AR-DNA interactions, reported to modulate the activator of the androgen receptor [22], were not assessed.

The residues within the wild-type DBD-DNA complex are highly dynamic, especially within the D-box region. The D-box loop is part of the Zn-finger motif, which is crucial for the interactions between AR monomers as well as for DNA-protein interactions. Such flexibility makes targeting the protein-protein interaction (PPI) interface of the AR-DBD by small molecules particularly challenging. Residues in the PPI interface fluctuated highly as a result of the simulations in the area comprised of charged components (i.e., DNA backbone and Zn^2+^) and polar residues, which further complicates the structure-based design of potent and selective small molecule inhibitors [22]. Pyrvinium pamoate was identified as the first small-molecule inhibitor of the AR-DBD; it is also among the first small molecules shown to directly inhibit AR splice activity; together with its variants it has important clinical implications in the treatment of CRPC [23]. Jones et al. showed through mutational analysis that it bound at the ARE-DBD interface, and residues from 609 to 612 were essential for pyrvinium’s mechanism of action [22]. Despite an interaction with the DBD-DNA complex, pyrvinium did not alter the DNA-binding kinetics [22]. More recently, Radaeva et al. reported two lead compounds, VPC-17160 and VPC-17281, identified in silico and followed up in cell-based assays, which inhibited AR by binding to the DBD domain and abrogated homodimerisation [22]. Compound VPC-17281 appeared to have favourable microsomal stability, which is important for future hit-to-lead optimisation, yet the affinity of the compounds needs to be improved. These promising proof-of-concept results show that small molecules can inhibit AR via binding to the DNA binding domain. Our simulations have shown that residues involved in the binding of those small molecules (e.g., T603, K610, R611) sampled a series of different configurational states through the nanoseconds to microseconds timescale, which is likely to contribute to the relatively low affinities of the compound (e.g., K_i_ within µM range reported for pyrvinium) [22]. Challenges of direct DBD targeting rendered the LBD a more attractive binding site for small molecules, given its reported allosteric sites and highly conserved orthosteric pocket [24,25,26,27]. However, AR mutants lacking LBD (e.g., ARV7; exon 4 deletion), which remain functional receptors yet insensitive to inhibitors targeting LBD, are the leading cause of drug resistance.

Several gain-of-function mutations have been found in the DBD and linked to prostate cancer pathology [15,16,17]. The K581R mutant was demonstrated to respond to non-androgen ligands (estradiol, progesterone, hydrocortisone, flutamide or bicalutamide), which caused elevated activation as a result of promiscuous binding [28]. The L575P-activating mutation was found in bicalutamide- and flutamide-treated castrated mice with primary prostate tumour and lymph node metastasis, but the specific role of this mutation is unknown [15,29]. A587V and A588S are somatic mutations found in prostate cancer, but, similarly to L575P, their specific roles remain unknown [30,31]. The only residue involved in direct interactions with DNA and/or the dimerisation interface is K581 (Figure 5), therefore, elucidating the exact role of L575P, A587V, A588S in AR activation is challenging.

We found that none of the four mutations studied in this work increased the monomer tertiary structure stability. However, each mutation showed a distinct configuration when bound to the ARE element. The L575P mutation did not affect the overall stability of the protein-DNA complex. However, burying deeper within the major groove, increasing the number of protein-DNA contact points and therefore strengthening interactions between AR and DNA, may explain the relationship between the L575P mutation and symptoms of castration-resistant prostate cancer (CRPC) [15]. The A587V mutation caused the structural change within the protein, as the steric clashes caused by bulkier valine residue affected the dimerisation interface. This rearranged the hydrogen-bonding network, driven by the interactions between Y594 and N611, which increased the interfacial stability of the homodimers [15,29].

Both polar mutations studied in this work (A588S and K581R) caused a significant shift in the homodimer configuration. Albeit less prominent in A588S, the twist movement sampled in both mutations indicated that the polar hydroxyl group of S588 directly affected the stability of the P-box, by destabilising the helix in which the P-box is located, and by changing its electrostatic environment. For the K581R mutation, the dimer’s disassembly occurred due to steric clashes between ARE nucleotide bases and R581, alongside the loss of the salt bridge located between the ARE backbone and K581. This resulted in weakening of the interactions between ARE and AR-DBD.

We focused not only on the impact of the mutations in the AR, but also on the possible effects exerted by somatic DNA mutations at the non-coding regions where DBD binds to DNA. Lack and coworkers showed that DBD binding sites have a dramatically increased rate of mutations that is greater than any other transcription factor and specific to prostate cancer [18], and provided evidence that these mutations are caused by impaired repair of abasic sites [19]. In addition to initiating tumour growth, there is also evidence that AR signalling is associated with DNA damage [32,33] and changes in autophagy [34].

Apurinic and apyrimidinic sites are the most commonly occurring DNA lesions in prostate cancer [35,36]. Alongside their altered structure and dynamics, abasic lesions are chemically unstable and prone to forming covalent crosslinks [37,38]. In normal cells, those lesions are efficiently repaired by the base excision repair (BER) mechanisms [39,40,41], yet abasic lesions seem to present a challenge in prostate cancer. This challenge comes from the increase in the formation of covalent DNA-protein crosslinks, due to impaired repair mechanisms [42]. The impact of losing a nucleobase on isolated B-DNA conformation and dynamics is well studied, but this has never been investigated in ARE research [43,44,45]. To date, several proteins, including apurinic/apyrimidinic endonuclease 1 (APE1) [46] and GAPDH [47,48], can interact with the deoxyribose of the abasic lesion to form a Schiff base via cysteine and lysine residues [49]. DBD is enriched in lysine residues in the interaction with DNA, which opens up a possibility that DBD could also form covalent crosslinks with abasic lesions in conditions such as prostate cancer.

Since abasic lesions within DNA were reported to affect the function of multiple DNA-binding proteins, and since AR binding sites are highly mutated in prostate cancer [18], we studied three different chemical moieties which can be found in abasic lesions: an esterfuranose, a hydroxyfuranose, and open furanose ring. Although the RMSD changes have similar magnitudes independent of the strand and the type of lesion, we found that lesions affected the conformations sampled by the protein. This, in combination with the observation that neither hydroxyfuranose nor esterfuranose lesions formed any specific interaction with the protein interface, can explain why the overall dynamics of the complex were not affected by the type of lesion, but the overall stability of the interface was. In the open-ring aldehyde, there was a specific interaction observed between the lesion and K610 side chain. This interaction was consistently sampled in several simulation replicas, but it required a subtle yet distinct conformational change to occur. This conformational change, evidenced by the higher protein RMSD values for the reverse-strand aldehyde, led to a more favourable interaction between protein and DNA. Collectively, the increase of DNA-protein interactions by abasic lesions by either stabilisation of the interaction interface, or specific interactions between lesion and K610 in the protein DBD domain, may suggest how such abasic lesions might activate the androgen receptor. The specific role of K610 will be elucidated in a subsequent study.

The quantum mechanical calculations indicated the plausibility of a reaction between the open-ring aldehyde and the K610. Given that the K610 is a positively charged residue, the LUMO orbital was located on top of the tertiary amine group within the side chain, and the HOMO orbital was located primarily on the aldehyde group of the lesion site. The last frontier bonding orbital was different from HOMO, and it was sparsely divided between the lesion and the K610 side chain. It is likely that K610 and the lesion may form a reversible covalent adduct via Schiff base formation, which would result in an activation of the AR receptor. Validation of the calculations presented in this work will require experimental studies aiming at detection of specific abasic lesions and putative DNA-protein adducts.

## 4. Materials and Methods

### 4.1. Molecular Dynamics Simulations

For all simulations in this work, the initial structure of the protein was a homology model of the DNA-binding domain of human androgen receptor (Uniprot accession number P10275). The template used to generate this model was the DNA binding domain of rat androgen binding receptor (PDB code: 1R4I) [50]. The sequence identity between human and rat DBD domain is 96%, with a 98% similarity. Given the high identity, the point mutations were modelled using UCSF Chimera [51] (https://www.cgl.ucsf.edu/chimera/, accessed on 31 October 2022).

All simulations were performed using GROMACS 2019 [52] (https://www.gromacs.org/, accessed on 31 October 2022). The equilibrium simulations were run in sets: (1) AR-DBD wild-type (WT) monomer, and four different point mutations: L575P, K581R, A587V, A588S; (2) AR-DBD WT dimer and four different point mutations: L575P, K581R, A587V, A588S; and (3) three abasic modification within the ARE element: open furanose ring, a closed furanose ring in a keto form, and a closed furanose ring in an eno form. Each abasic substitution was made in two positions: adenosine 11 of the leading strand, and adenosine 28 of the reverse strand.

All three sets were parametrised using the AMBER99SB-ILDN force field and immersed in the cubic box of TIP3P [53] water model. Non-standard residues were parametrised using GAFF [54] via ACPYPE [55] and added to the force field database within GROMACS. Box distance was set to 1 nm from the edge of the protein, and standard 3D periodic boundary conditions (PBC) were applied. The box was solvated, and Na^+^ and Cl^−^ ions were added to achieve a 0.1 M/L concentration, and to maintain the charge neutrality of the simulation box. The solvated systems were energy-minimised and equilibrated. The minimisation ran using steepest descent for 1000 cycles, followed by the conjugate gradient minimisation. The energy step size was set to 0.001 nm, and the maximum number of steps was set to 50,000. The minimisation was stopped when the maximum force fell below 1000 kJ/mol/nm using the Verlet cutoff scheme. Treatment of long-range electrostatic interactions was set to Particle Mesh-Ewald (PME) [56], and the short-range electrostatic and van der Waals cutoff was set to 1.0 nm. After the energy minimisation, heating to 300 K was performed for 20 ps with a time step of 2 fs and position restraints applied to the backbone in an NVT ensemble. The constraint algorithm used was LINCS (Linear Constraint Solver) [57], which was applied to all protein and DNA bonds and angles. With the Verlet cut-off scheme and the non-bonded short-range interaction, the cut-off was set to 1.0 nm. Long-range electrostatics were again set to PME. The temperature coupling was set between the protein and the non-protein entities by using a Berendsen thermostat, with a time constant of 0.1 ps, and the temperature set to reach 300 K with the pressure coupling turned off. Pressure equilibration was run at 300 K with the Parrinello-Rahman pressure coupling on, and set to 1 bar [58] in an NPT ensemble. The equilibrated trajectories were set to three replicates of 300 ns for dimer and dimer + DNA, and 500 ns for monomer systems, with the first 10 ns discarded from the analysis.

Analysis of the trajectories was performed using GROMACS tools, including root-mean-square deviation (RMSD) to assess overall stability, and per-residue root-mean-square fluctuation (RMSF) to assess the local flexibility. Hydrogen bond analysis was carried out using VMD [59].

### 4.2. Quantum Mechanical Calculations

The C11 5′-3′ molecular orbital calculations were performed using Gaussian09 [60]. The average configuration obtained from all-atom molecular dynamics simulations was prepared in Gaussview [61]. The residues of interest were separated from the rest of the molecules for simplicity, and position restraints were added to the remaining backbone atoms. The geometry has been optimised using DFT level of theory, with B3LYP functional and a 6-311G* basis set, using PCM with water model for implicit solvation.

## 5. Conclusions

In this work, we used all-atom molecular dynamics simulations to investigate the effects of a series of oncogenic point mutations in the androgen receptor DNA binding domain, and a series of abasic lesions in its respective ARE element. Our simulations investigated how, mechanistically, those CRPC-linked mutations affected the structure, dynamics and interactions within ARE-DBD complexes. It seems that the AR-activating effects of these mutations arose from the conformational changes within the protein and the weakening of the interaction network between ARE and AR-DBD. Regarding the effects caused by abasic lesions within the DNA strands, our data have shown the possibility of the formation of reversible covalent adducts between DNA and protein, which would increase the protein-DNA interactions, prolong the lifetime of the complex, and therefore activate the expression of AR-controlled genes. We expect that this work will help in guiding future experiments aimed at detection of specific adducts. This could be a starting point of a structure-guided drug discovery campaign, addressing castration-resistant prostate cancer refractory to treatment.

In addition, future work, which will build upon the results of this study, will investigate the impact of specific mutations (in DNA and DBD) on AR “druggability” by small molecules and develop ligands that block the AR-DBD dimerisation interface in those mutated variants—an attractive target given its role in AR activation and independence from the LBD. Such inhibitors may potentially circumvent LBD-dependent resistance mechanisms and directly target CRPC currently refractory to therapeutics.

## Figures and Tables

**Figure 1 ijms-24-01270-f001:**
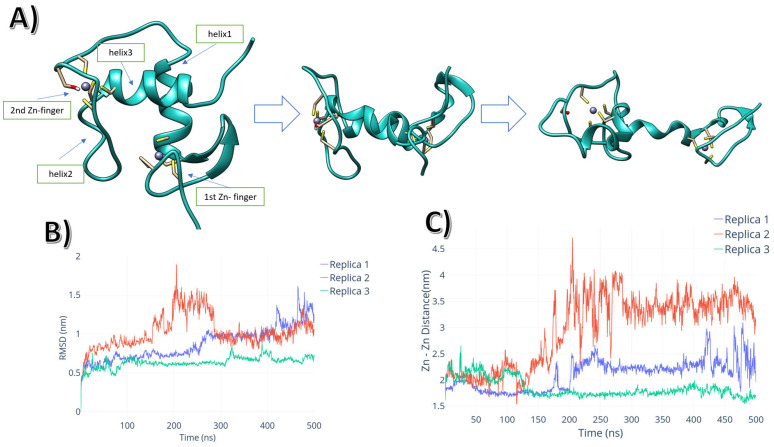
(**A**) AR-DBD monomer unfolding pathway. (**B**) RMSD plot for each of the monomer replicas vs. time. (**C**) Inter Zn^2+^ distance vs. time.

**Figure 2 ijms-24-01270-f002:**
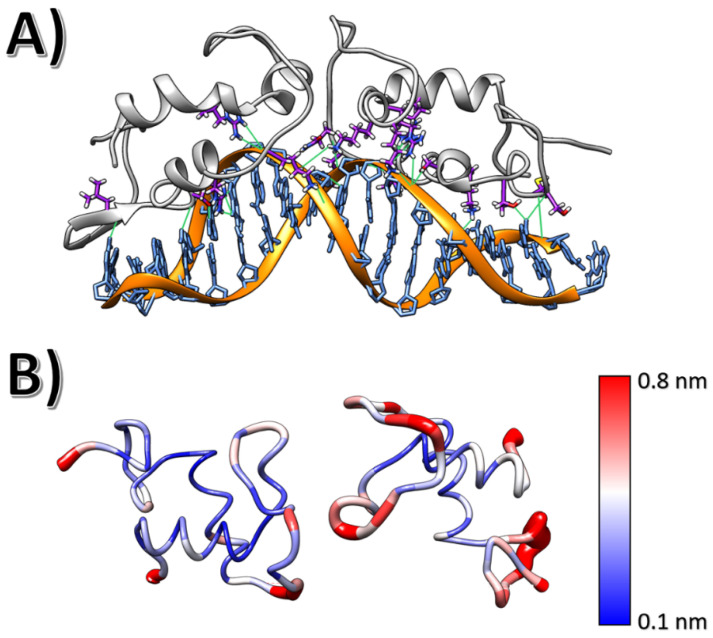
(**A**) AR-DBD-ARE average conformation acquired during a 300 ns MD simulation. AR-DBD residues were found to interact with the ARE element; hydrogen bonds are depicted in green. (**B**) Root mean square fluctuation per residue for both AR-DBD monomers, with the respective colour scale.

**Figure 3 ijms-24-01270-f003:**
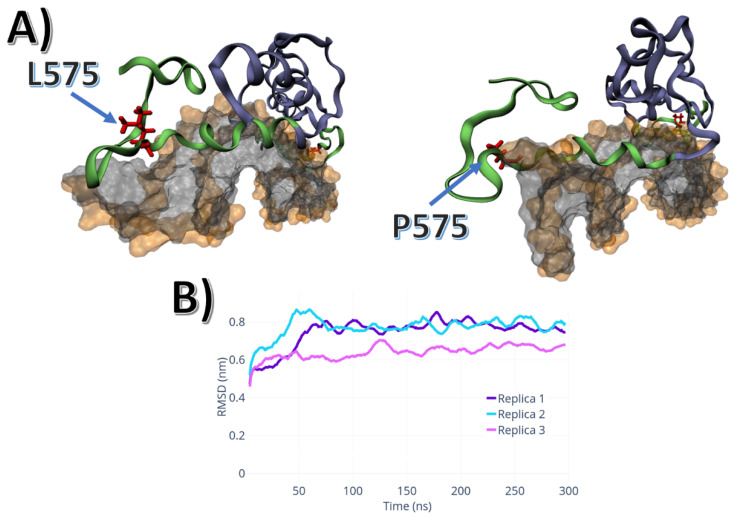
L575P mutation dynamics: (**A**) Conformational differences between L575 and P575: the last 20 residues in the C-terminus, which comprises the P-Box and helix 2. (**B**) RMSD of the three P575 replicas.

**Figure 4 ijms-24-01270-f004:**
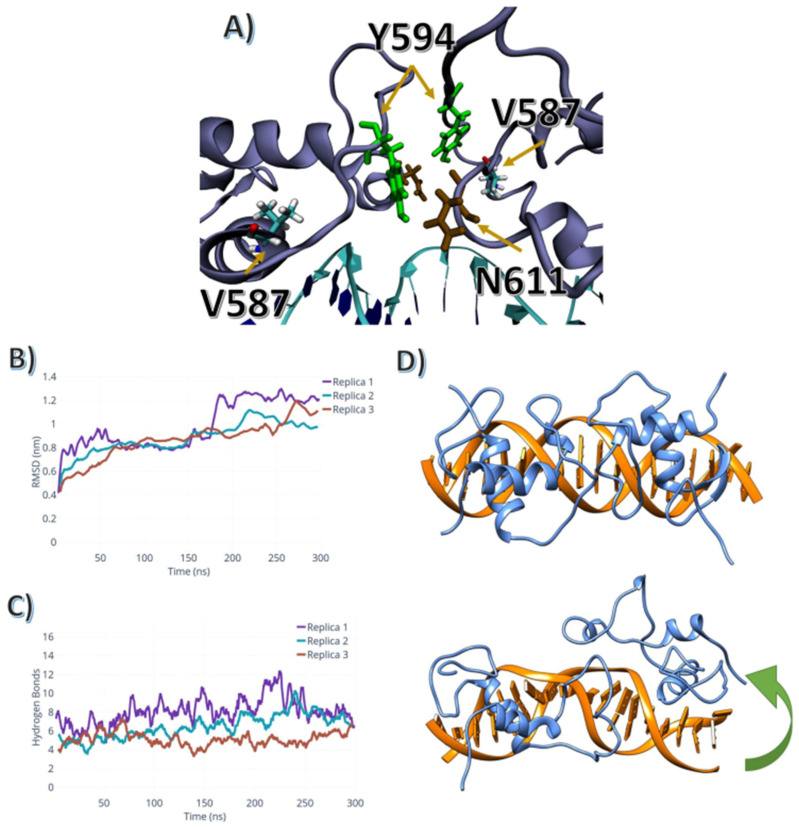
V587 and A588S mutation conformational changes: (**A**) V587 affects the dynamics of both N611 (brown) and Y594 (green). (**B**) RMSD for the A588S mutation simulations. (**C**) Total number of hydrogen bonds between the ARE and the AR-DBD for the A588S mutant. (**D**) The A588S mutant twist movement; the AR-DBD (blue) and the DNA-ARE (orange); The green arrow indicates the twist motion which the homodimer containing the A588S mutation undergoes.

**Figure 5 ijms-24-01270-f005:**
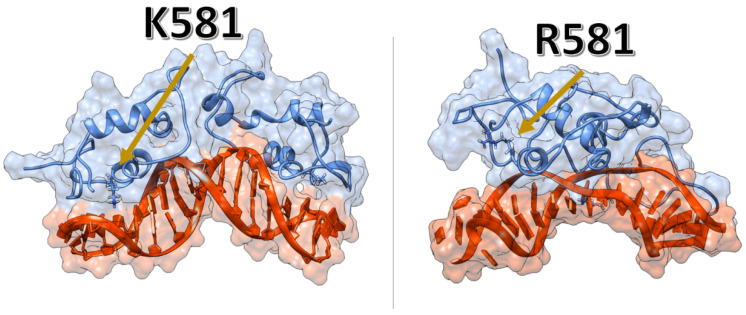
K581R mutant weakens the interactions between the protein and the DNA-ARE. Left panel—the average configuration sampled from the WT simulations. Right panel—the pocket formed between the AR-DBD and the DNA-ARE, wherein R581 is located.

**Figure 6 ijms-24-01270-f006:**
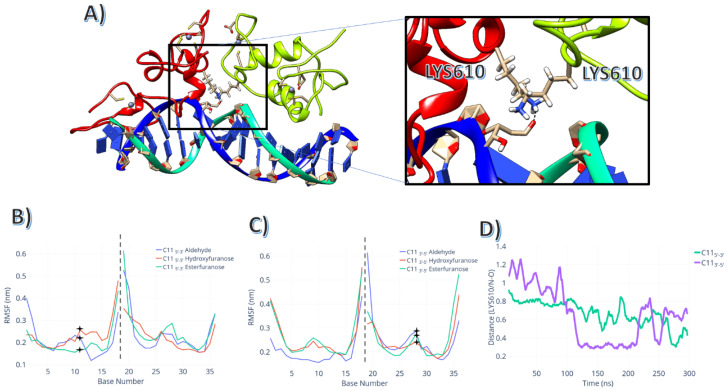
Intrinsic dynamics of the complex with C11 abasic lesions: (**A**) Aldehyde-formed interaction with K610. (**B**) RMSF for the DNA chains for the simulations with the abasic lesion located in the 5′-3′ strand; stars indicate the abasic lesion’s position. (**C**) RMSF for the DNA chains for the simulations with the abasic lesion located in the 3′-5′ strand; crosses indicate the abasic lesion’s positions. (**D**) distance between the NZ atom (side chain amine group) and the oxygen atom of the aldehyde group in the abasic lesion.

**Figure 7 ijms-24-01270-f007:**
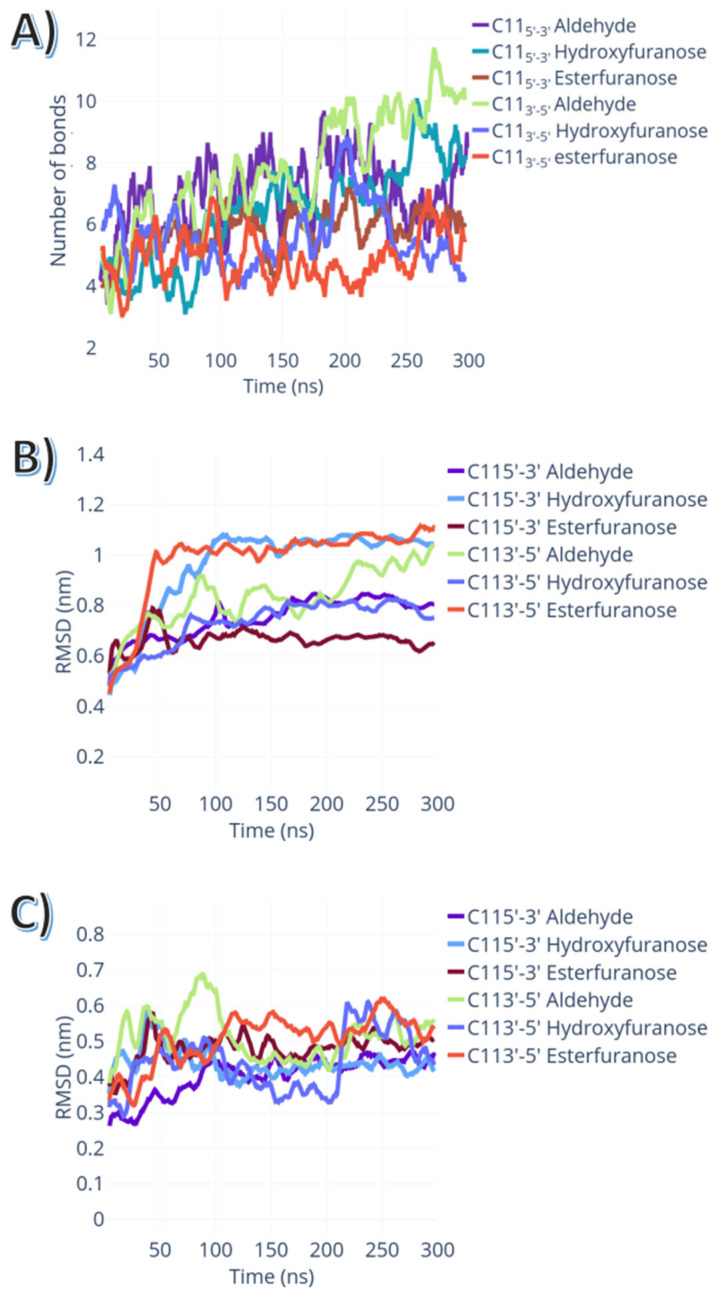
Impact of different abasic lesions on the intrinsic dynamics of the AR-DNA complexes. (**A**) the total number of hydrogen bonds between AR-DBD domain and the lesioned ARE element. (**B**) Protein RMSD in relation to the whole ternary complex structure vs. simulation time. (**C**) DNA RMSD in relation to the whole ternary complex structure vs. simulation time.

**Figure 8 ijms-24-01270-f008:**
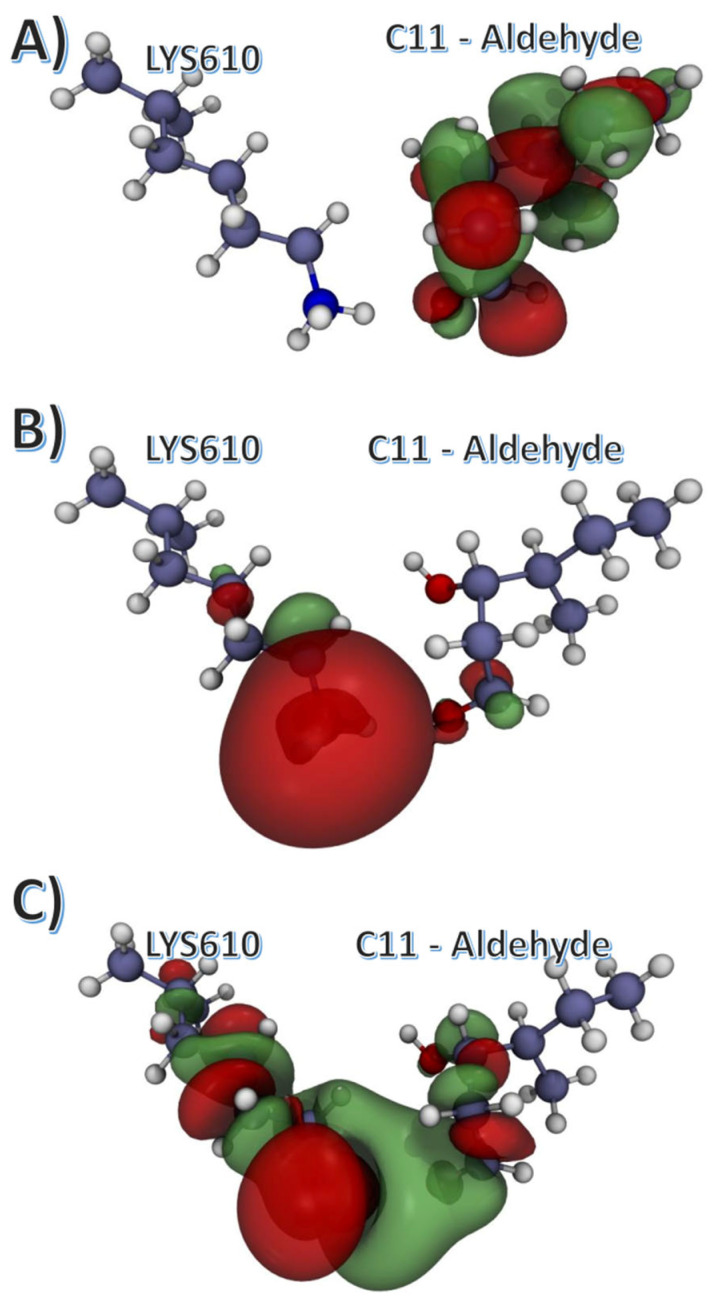
Calculated molecular orbitals for the average configuration of the C11 3′-5′ lesion. Aldehyde configuration with K610. (**A**) Highest-occupied molecular orbital (HOMO); (**B**) Lowest-unoccupied molecular orbital (LUMO); (**C**) Lower frontier orbital.

## Data Availability

All data generated for this study, which is not included within the manuscript or associated files (Appendix A), is available upon request.

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
