# Peer review of "Mechanistic Investigation of the Androgen Receptor DNA-Binding Domain and Modulation via Direct Interactions with DNA Abasic Sites: Understanding the Mechanisms Involved in Castration-Resistant Prostate Cancer"

_ijms, 2023, doi:10.3390/ijms24021270_

Round 1

Reviewer 1 Report

In the paper by Xu et al. titled "Mechanistic investigation of the androgen receptor DNA-binding domain and modulation via direct interactions with DNA abasic sites: understanding the mechanisms involved in castration-resistant prostate cancer" the authors conducted an extensive in silico analysis of androgen receptor DBD-DNA interactions. Specifically they focused the stability of the non-DNA bound dimer, several gain of function mutations (L575P, K581R, A587V, and A588S ) and binding to abasic DNA. Overall the study was well done and is of interest to the field. While outside this project, experimental validation would be extremely helpful to strengthen the impact of this work particularly in regards to some of the authors more controversial claims.   General comments:
  • I would like to see greater discussion on the potential effect of these DBD mutations on DNA binding. Specifically, what can these MD simulations tell us about how they could potentially influence AR binding or activity. 
  Specific comments: 
  • Pg 1, Line 15-16: Nearly all AR variants do not contain a LBD.
  • Pg 2, Line 50: Please change "characterized by AR signalling return" to "characterized by the return of AR signalling"
  • Pg 2, Line 66-67: Cannot tell from ChIPseq if the binding is direct (DBD-mediated) or indirect (PPI-mediated).
  • Pg 2, Line 90: I believe Morova et al. demonstrated that AR binding sites had increased mutations in primary PCa not CRPC.
  • Pg 4, Line 140: Give timescale used to model the AR-DBD-ARE average confirmation in figure legend
  • Pg 11, Line 303: silico is misspelled.
  • Pg 11, Line 306: tye is misspelled.
  • Pg 12, Line 352: The authors state that "(this) specific interactions between lesion and K610 in the protein DBD domain, may explain how such abasic lesions could activate the androgen receptor". However I don't believe there is any experimental data to suppor this concept. Please give specific reference or remove. 

Author Response

On behalf of my coworkers and myself, I would like to thank the Reviewer for reading our manuscript (ID: ijms-2100124) and providing very constructive comments, which we have addressed. All the corrections suggested by the Reviewer were applied through the revised manuscript; the edited sections are highlighted. In addition, the number of the joint first authors has been reduced to two to comply with the journal guidelines, and the order of the authors has been adjusted accordingly.

Responses and clarifications to specific comments are given below:

  • I would like to see greater discussion on the potential effect of these DBD mutations on DNA binding. Specifically, what can these MD simulations tell us about how they could potentially influence AR binding or activity. 

The section in the Discussion has been restructured, and the following paragraph has been added:

The K581R mutant was demonstrated to respond to non-androgen ligands (estradiol, progesterone, hydrocortisone, flutamide or bicalutamide), which caused elevated activation caused by promiscuous binding [18]. The L575P activating mutation was found in bicalutamide- and flutamide-treated castrated mice with primary prostate tumour and lymph node metastasis, but the specific role of this mutation is unknown [15,19]. A587V and A588S are somatic mutations found in prostate cancer, but, similarly to L575P, their specific roles remain unknown [20,21]. The only residue involved in direct interactions with DNA and/or dimerisation interface is K581 (Figure 5), therefore elucidating exact role of L575P, A587V, A588S in AR activation is challenging.

We found that none of the four mutations studied in this work increased the monomer tertiary structure stability. However, each mutation showed a distinct configuration when bound to the ARE element. The L575P mutation did not affect the overall stability of the protein-DNA complex. However, burying deeper within the major groove, increasing the number of protein-DNA contact points and therefore strengthening interactions between AR and DNA, may explain the relationship between L575P mutation and symptoms of castration-resistant prostate cancer (CRPC) [49]. The A587V mutation caused the structural change within the protein, as the steric clashes caused by bulkier valine residue affected the dimerisation interface. This rearranged the hydrogen-bonding network, driven by the interactions between Y594 and N611, which increased the interfacial stability of the homodimers [15,19].

Both polar mutations studied in this work (A588S and K581R) caused a significant shift in the homodimer configuration. Albeit less prominent in A588S, the twist movement sampled in both mutations indicated that the polar hydroxyl group of S588 directly affected the stability of the P-box, by destabilising the helix in which P-box is located, and by changing its electrostatic environment. For the K581R mutation, the dimer's disassembly occurred due to steric clashes between ARE nucleotide bases and R581, alongside the loss of the salt bridge located between the ARE backbone and K581. This resulted in weakening of the interactions between ARE and AR-DBD.

  • Pg 1, Line 15-16: Nearly all AR variants do not contain a LBD.

Addressed  by adding “nearly all AR” to p. 1, l. 16

  • Pg 2, Line 50: Please change "characterized by AR signalling return" to "characterized by the return of AR signalling"

Addressed by changing the original sentence to “characterised by the return of AR signalling”

  • Pg 2, Line 66-67: Cannot tell from ChIPseq if the binding is direct (DBD-mediated) or indirect (PPI-mediated).

The part of the text has been edited, and the sentence “However, the impact of those abasic lesions on structure and dynamics of AR-DNA complex has not been studied to date.” Has been added to the end of the paragraph.

  • Pg 2, Line 90: I believe Morova et al. demonstrated that AR binding sites had increased mutations in primary PCa not CRPC.

The sentence has been restructured, and it now reads as “ ”.

  • Pg 4, Line 140: Give timescale used to model the AR-DBD-ARE average confirmation in figure legend

Done by adding “ acquired from a 300 ns MD simulation”

  • Pg 11, Line 303: silico is misspelled.

Addressed by changing sílico into “sillico”.

  • Pg 11, Line 306: tye is misspelled.

Addressed by change tye into “the”.

  • Pg 12, Line 352: The authors state that "(this) specific interactions between lesion and K610 in the protein DBD domain, may explain how such abasic lesions could activate the androgen receptor". However I don't believe there is any experimental data to suppor this concept. Please give specific reference or remove. 

There is no experimental data available yet for this specific K610 residue within human androgen receptor. However, similar mechanism has been reported for APE1 by Ilina and coworkers (Biochimie 2018 Jul;150:88-99. doi: 10.1016/j.biochi.2018.04.027), with in vitro and in vivo experimental data showing the presence of crosslinks between lysine residues within N-terminal domain of APE1 and abasic lesions. Therefore, the mechanism has precedence and the citation has been added. Our follow-up work, which we are conducting at the moment in collaboration with experimental molecular oncology group in Newcastle (Translational and Clinical Research Institute Newcastle University Centre for Cancer) aims at experimental validation of the modelling outcomes reported in this paper, and we expect to submit the results of the follow-up study as soon as possible.

We have added the paragraph to the Discussion, providing the context, edited the sentence highlighted by the Reviewer, and added the reference to Ilina’s paper (ref. 60).

Reviewer 2 Report

The authors used all-atom molecular dynamics simulations to investigate the effects of a series of oncogenic point mutations in the androgen receptor DNA binding domain, and a series of basic lesions in its respective ARE element. The topic is of considerable interest and well described. The article structure must be improved. Images are clear and well-drawn. A major revision is required.

-        Introduction: lines 53-87. I believe this part is out of the theme for an introduction. However, I have appreciated the content and I suggest reconsidering shifting this part in the discussion section.

-        Section 5. Why did you discuss materials and methods after the conclusion? Please consider reorganizing a more established structure of the manuscript. 

-        Autophagy plays a crucial role in cancer development and in response to drugs. It plays a pivotal role in hormonal and chemotherapy resistance, and targeting its activity still represents a valid option to improve therapy effectiveness. I believe this is worthy of interest and I suggest discussing and including this in your paper. For this purpose, current evidence has investigated how autophagy interacts within these complex interactions. I suggest including one of the latest papers on the topic (doi: 10.3390/ijms23073826; PMCID: PMC8999129; PMID: 35409187). 

-        When presenting the conclusion be more concise. Avoid repetitions or introductions. Report only the evidence of your study and prospects.

-        Check typos.

Author Response

On behalf of my coworkers and myself, I would like to thank the reviewer for reading our manuscript (ID: ijms-2100124) and providing very constructive comments, which we have addressed. All the corrections suggested by the reviewer were applied through the revised manuscript; the edited sections are highlighted. In addition, the number of the joint first authors has been reduced to two to comply with the journal guidelines, and the order of the authors has been adjusted accordingly.

Responses to the specific comments are given below:

-        Introduction: lines 53-87. I believe this part is out of the theme for an introduction. However, I have appreciated the content and I suggest reconsidering shifting this part in the discussion section.

We appreciate this suggestion, however some of the content is required for providing the context and needs to be included in the introduction. We moved this section (l. 53-87) to the Discussion, following the Reviewer’s suggestion, and the introduction has the following, more concise paragraph instead:

To date, several gain-of-function mutations have been found in the DBD and linked to prostate cancer, such as K581R [15–18]. Other mutations, such as L575P, A587V and A588S, were found in prostate cancer, but their specific roles remain unknown [20-21]. Except K581, those residues are located far from the dimerisation interface, are not involved in direct interactions with DNA, and their exact role is yet to be elucidated. While mutations in the AR, particularly within the LBD domain, have been extensively studied in past years, little has been reported about somatic DNA mutations at the non-coding regions where DBD binds to DNA. Lack and coworkers have addressed this gap, employing clinical whole-genome sequencing [22]. They showed that DBD binding sites have a dramatically increased rate of mutations that is greater than any other transcription factor and specific to prostate cancer [22]. They also provided evidence that these mutations at DBD-DNA binding sites are caused by impaired repair of abasic sites [23]. Most AR-DBD-binding sites are in intronic or intergenic regions. Many of them contain an androgen response element (AREs) that consists of a 15-bp palindromic sequence 5′-AGAACA-NNN-TGTTCT-3′ or two hexameric direct repeat sequences 5’-AGAACA-NNN-AGAACA-3' with a 3 bp spacer [24]. In addition to initiating tumour growth, there is also evidence that AR signalling is associated with DNA damage [25,26]. However, impact of those abasic lesions on structure and dynamics of AR-DNA complex has not been studied to date.

-        Section 5. Why did you discuss materials and methods after the conclusion? Please consider reorganizing a more established structure of the manuscript. 

We have followed the IJMS template which we used previously, with the order of the sections is as follows: Abstract, Introduction, Results, Discussion, Conclusions, Materials and Methods, References. However, we have noticed that several other papers published recently have Conclusions after Materials and Methods, and instruction for authors permit placing Conclusions after Materials and Methods. Therefore, we have changed the order following the Reviewer’s suggestion. Thank you very much.

-        Autophagy plays a crucial role in cancer development and in response to drugs. It plays a pivotal role in hormonal and chemotherapy resistance, and targeting its activity still represents a valid option to improve therapy effectiveness. I believe this is worthy of interest and I suggest discussing and including this in your paper. For this purpose, current evidence has investigated how autophagy interacts within these complex interactions. I suggest including one of the latest papers on the topic (doi: 10.3390/ijms23073826; PMCID: PMC8999129; PMID: 35409187). 

This is a very interesting comment – thank you very much. We have added a paragraph in the Discussion, and cited the review paper suggested by the Reviewer (ref. 61).

-        When presenting the conclusion be more concise. Avoid repetitions or introductions. Report only the evidence of your study and prospects.

The Conclusions have been restructured.

-        Check typos.

These have been fixed.

We hope that our response thoroughly address all the concerns, and we are looking forward to the feedback.

Round 2

Reviewer 2 Report

The new version has been deeply improved. I appreciated the renovation of the paper and all the improvements. I am satisfied with this new version.